# Antimicrobial and Antibiotic Resistance from the Perspective of Polish Veterinary Students: An Inter-University Study

**DOI:** 10.3390/antibiotics11010115

**Published:** 2022-01-17

**Authors:** Tomasz Sobierajski, Beata Mazińska, Wioleta Chajęcka-Wierzchowska, Marcin Śmiałek, Waleria Hryniewicz

**Affiliations:** 1Faculty of Applied Social Sciences and Resocialization, Warsaw University, Krakowskie Przedmieście 26/28, 00-927 Warsaw, Poland; 2Department of Epidemiology and Clinical Microbiology, National Medicines Institute, Chełmska 30/34, 00-725 Warsaw, Poland; b.mazinska@nil.gov.pl (B.M.); w.hryniewicz@nil.gov.pl (W.H.); 3Department of Industrial and Food Microbiology, Faculty of Food Science, University of Warmia and Mazury in Olsztyn, Plac Cieszyński 1, 10-726 Olsztyn, Poland; wioleta.chajecka@uwm.edu.pl; 4Department of Poultry Diseases, Faculty of Veterinary Medicine, University of Warmia and Mazury, 10-719 Olsztyn, Poland; marcin.smialek@uwm.edu.pl

**Keywords:** antibiotics, veterinary, veterinary students, education, AMR

## Abstract

The phenomenon of antibiotic resistance is a global problem that affects the use of antibiotics by humans and animal husbandry. One of the primary reasons for the growing phenomenon of antibiotic resistance is the over-prescription of antibiotics by doctors in human medicine and the overuse of antibiotics in industrial animal farming. Adequate education of veterinary medical students on the use of antibiotics in animal husbandry may reduce antibiotic resistance. For this reason, a survey was conducted among students at four primary research and didactic centers teaching veterinary medicine in Poland. The survey aimed to find out the knowledge and attitude of students towards the use of antibiotics and antibiotic resistance. The survey was conducted in May/June 2021. Four hundred and sixty-seven students participated in the study. The study positively verified that antibiotics and antibiotic resistance knowledge increase with successive years of veterinary studies/education. For most students (82.2%), antibiotic resistance is a significant problem, but only 58.7% believe it is global, and one in three respondents heard about the One Health approach.

## 1. Introduction

Since their discovery, antibiotics have been broadly used to treat infections in humans and veterinary medicine. Moreover, they have been used to treat animal infections and, as growth promoters, to enhance production performance in the agriculture sector [1]. However, after many years of intensive use of antibiotics in different fields of our life, antimicrobial resistance has become one of the most significant threats to the public health. It diminishes the effectiveness of treatment of infections not only in humans but also in animals [2,3,4,5,6]. A strong association between antibiotic use and resistance has been established [7,8,9]. The more we use, the faster resistant strains emerge and spread, and they may be transferred from humans to animals and vice versa. Antibiotic resistance is a dynamic process involving gene exchange between human and animal pathogens [10,11,12,13,14]. It can take place directly through the food chain or environment. European countries have early realized that antibiotics as growth promotors constitute an excellent risk for developing and spreading resistant bacteria [15,16,17]. Consequently, on 1 January 2007, a ban was declared. It took ten more years for the USA to declare the same ban. The World Health Organization (WHO) and several public health agencies warn that if we do not take immediate multifactorial actions to delay resistance development and spread, the consequences will be overwhelming and most likely irreversible for human and animal health, the economy, and the environment. WHO developed a so-called One Health approach, embracing people and animals closely connected within the shared environment [18,19,20].

One of the critical actions required to combat resistance is to increase knowledge regarding the causes and consequences of the overuse and misuse of antibiotics, increasing awareness of antibiotic resistance [21]. Veterinarians belong to a group of professionals diagnosing infections and prescribing antibiotics. Therefore, they should be up to date with the knowledge on antibiotic treatment, resistance, and when antibiotics are of real benefit [22]. Previous studies have evaluated and analyzed the attitude and knowledge of different medical professionals, including medical and dentistry students, on antibiotics and their use [23,24]. Several gaps in their knowledge were demonstrated, which should be remediated with further education. Since veterinarians play an important role in antibiotic usage, in the current study, we analyzed how veterinary students at different levels of education perceive the problem of antibiotic prescribing and resistance.

## 2. Results

### 2.1. Study Participants

A total of 467 veterinary medicine students from four academic centers participated in the study. Student participation by the academic year was roughly equivalent, except for 6th year, when the number of full-time classes is minimal, and students are mainly involved in clinical rotations. Most of the respondents were female (*n* = 358, 76.7%), which corresponds to the gender-demographic distribution of veterinary students in Poland. The study group was homogeneous in terms of ethnicity and diverse in terms of provenance. Most respondents came from urban areas (*n* = 315, 67.5%), and one in three respondents came from rural areas (*n* = 152, 32.5%). The selected characteristics of the study group are shown in Table 1.

More than one-third of students (*n* = 164, 35.1%) had taken antibiotics in the past year, and 25.3% (*n* = 118) had taken antibiotics 1–2 years ago. The vast majority of students (*n* = 390, 93.1%) used an antibiotic prescribed by a physician or dentist, 4% received an antibiotic from a family member/friend or used leftovers from previous treatment, 1% used an antibiotic prescribed by a veterinarian, and 1% purchased an over-the-counter antibiotic from a pharmacy. Most respondents used the prescribed antibiotic as recommended by their physician (*n* = 392, 93.6%) (Table 2).

### 2.2. Influence of Educational Programs on Students’ Attitudes toward Antibiotic Use

Most survey participants (*n* = 332, 71.1%) rated their knowledge of antibiotics positively/adequately, while 28.9% (*n* = 135) as poor (Table 2). These were more often students I-III (*n* = 88, 65%) than students in years IV–VI (*n* = 47, 35%).

The vast majority of respondents (*n* = 418, 89.5%) indicated that they learned about the growing problem of antibiotic resistance during their studies. In this group of students, 73.9% (*n* = 309) rated their knowledge rather positively/as adequate and 26.1% (*n* = 109) rather inadequate/negatively.

Seven out of ten students (*n* = 325, 69.6%) indicated that participation in classes (veterinary college classes) influenced/motivated/inspired them to learn more about antibiotic use in humans (Table 3). Most students in this group (79.4%) rated their knowledge (positively) as adequate and 20.6% as inadequate (negatively). Among the respondents who said that the classes at the university increased their understanding of the use of antibiotics in humans, the largest group were residents of big cities—29%, followed by small towns—25%, medium-sized cities—18%, villages without farms—18% and the smallest group of people from villages with farms—10%. Responses that the classes did not increase knowledge of antibiotic use were more often given by students in years I–III (*n* = 91, 64%) than in years IV–VI (*n* = 51, 36%) (*p* < 0.001).

Most respondents (*n* = 375, 80.3%) indicated that (veterinary college) participation in classes influenced them to learn about animal antibiotic use (Table 3). In this group of students, 78.4% rated their knowledge positively/as adequate and 21.6% as inadequate (negatively). One-fifth of the students (*n* = 92, 19.7%) stated that the knowledge imparted/acquired at the university did not influence their knowledge of the use of antibiotics in animals. This was more common among students in years I–III (*n* = 63, 68%) than IV–VI (*n* = 29, 32%) (*p* < 0.001). Most respondents (*n* = 354, 75.8%) indicated that the knowledge gained in college had no effect on negating the therapy ordered by a veterinarian for their own or their friends’ pet’s illness (Table 3). Among this group, 69.8% of students were confident about their knowledge of antibiotics.

Most respondents (*n* = 364, 77.9%) declared that they had not encountered a situation in which a veterinarian prescribed an antibiotic for personal use (Table 4). Students from years IV–VI were more likely to experience this situation (*n* = 63, 61%) than students of years I–III (*n* = 40, 39%) (*p* < 0.001). Half of the respondents (*n* = 242, 51.8%) would self-prescribe an antibiotic for personal use *ad usum prioprium* (Table 4), whereas 48.2% (*n* = 225) of survey participants would not do so. Respondents representing older (IV–VI) years were more likely to declare that they would self-prescribe an antibiotic for personal use (*n* = 135, 56%) than respondents of younger (I–III) years (*n* = 107, 44%) (*p* < 0.001).

### 2.3. Practical Knowledge of Antibiotics

Almost all respondents knew that antibiotics are ineffective for treating infections caused by viruses (*n* = 451, 96.6%) and are effective against bacteria (*n* = 438, 93.8%). They also are aware that “improper use of antibiotics can cause microorganisms to become resistant” (*n* = 445, 95.3%). Most students (*n* = 362, 77.5%) disagree that “using antibiotics will make people resistant to them”, and 22% of the respondents agreed with this statement. In this group, 57% were students from years I–III and 43% from IV–VI (*p* = 0.005). More than half of the respondents (*n* = 257, 55%) expressed that the “use of antibiotics often cause side effects”. At the same time, 45% of students disagreed with this statement. This opinion was expressed slightly more often by students in years I–III (*n* = 116, 55%) than IV–VI (*n* = 94, 45%) (*p* = 0.257). Most students (*n* = 397, 85%) agreed with the statement that “bacteria pass information about antibiotic resistance to each other” (Medians of students’ knowledge of antibiotics included in Table 5).

Nearly all respondents (*n* = 460, 98.5%) would like to increase their knowledge regarding antibiotic use in animals. Six in ten respondents (*n* = 278, 59.5%) said they would check antibiotic information and recommendations in another source before starting their doctor-ordered treatment. One in three (*n* = 168, 36%) would start the treatment fully trusting the doctor.

Half of the respondents (*n* = 228, 48.8%) would check antibiotic information from another source before starting a treatment ordered for their pet by a veterinarian.

Among those who gave the above response, 50% represented years I–III and 50% years IV–VI (*p* < 0.001). At the same time, 45% of respondents would start the treatment fully trusting the doctor. This opinion was expressed by more students in years I–III (*n* = 129, 65%) than students IV–VI (*n* = 69, 36%) (*p* < 0.001).

Students declared that they would seek information about the use of an antibiotic when treating their pet or the pet of someone close to them: in the drug leaflet (*n* = 419, 89.7–60.4% “definitely yes” responses), from a veterinarian (*n* = 405, 86. 9–48% of “definitely yes” responses), on the Internet (*n* = 407, 87.2–49.7% of “definitely yes” responses), and in hard copy literature (*n* = 331, 70.9–31.5% of “definitely yes” responses).

### 2.4. Knowledge of the Phenomenon of Antibiotic Resistance

Most students (*n* = 384, 82.2%) believe that the issue of antibiotic resistance is a significant problem. At the same time, 17% of the students think it will only become a problem in the future. Most students perceive the problem of antibiotic resistance at a global level (*n* = 274, 58.7%), one in nine at an EU level (11%), and one in eight at a national level (12%). One in eight respondents (12%) answered “don’t know/difficult to say”.

One in three respondents (*n* = 169, 36.2%) had heard of the One Health approach. On average, one in seven students had heard of the National Antibiotic Program (*n* = 69, 14.8%) and European Antibiotic Awareness Day (*n* = 60, 12.8%).

According to the students, low awareness of the dangers of antibiotic resistance (*n* = 235, 50.3% for the “strongly agree” response) and the overuse of antibiotics by physicians (*n* = 182, 39% for the “strongly agree” response) have the most significant impact on the growing phenomenon of antibiotic resistance. In contrast, the use of antibiotics without a prescription (self-medication) has the most negligible impact (*n* = 77, 16.5% for “strongly disagree” responses) as well as the overuse of antibiotics by dentists (*n* = 62, 13.3% for “strongly disagree” answers) (Table 6).

Most students (*n* = 354, 75.8%) believe there are current recommendations for using antibiotics in specific clinical situations for animal species in veterinary medicine. At the same time, one-fifth (*n* = 100, 21.4%) do not know these recommendations. Of this group, 77% were students in years I–III and 23% in years IV–VI (*p* < 0.001).

## 3. Discussion

The knowledge, attitude, and awareness towards the antibiotic resistance of veterinary students can lead to better combating of antimicrobial resistance in the future. Our study indicates that veterinary colleges in Poland prepare students well in expanding their knowledge of antibiotics.

Students of veterinary medicine in Poland (irrespective of their year of study) are aware of the increasing antibiotic resistance. They are familiar with the general mechanisms of antibiotic action and their spectrum of activity. It is evidenced by the fact that 97% of respondents negated the effectiveness of antibiotics against viruses and 85% of responses agreed with the statement that bacteria can pass information about antibiotic resistance among themselves. Polish veterinary students see as the main reason for the rise in antibiotic resistance the low awareness of the dangers of increasing antibiotic resistance due to their overuse in both human and veterinary medicine. Interestingly, all year groups believe that the overuse of antibiotics by physicians is more influential for the rise in antibiotic resistance than overuse in veterinary medicine. According to the students, limited access to microbiological diagnostics and low levels of hygiene in animal husbandry is also crucial in this respect. Data from scientific literature show that the unprecedented increase in antimicrobial-resistant organisms is linked to the massive use of antimicrobial agents for infection treatment and prevention [25,26,27]. This practice facilitates the emergence of resistant strains, and they spread at the farm level and beyond [28]. However, this phenomenon is determined not only by these differences. As it turns out, a high correlation has been demonstrated between the increased incidence of MDR E. coli in food products from poultry and the practices and procedures applied in a processing plant manufacturing these products [25]. This also allows us to conclude that the course of studies conducted at the faculties of veterinary medicine in Poland covers the phenomenon of antibiotic resistance quite reasonably. However, there are still many elements that would need to be strengthened in the learning process, with particular attention to the global nature of AMR.

Patient attitudes also contribute to inappropriate and excessive antibiotic use. More than half (60.4%) of the students in our study had taken an antibiotic in the past 24 months, and more than one-third had taken antibiotics in the past year. This result is higher than the answers given by Polish respondents participating in the Eurobarometer 2018 survey, where 24% had taken antibiotics in the last year [29]. On the other hand, in a study conducted in 2018 among medical students at the Warsaw Medical University, almost half (46%) had taken antibiotics in the last year [23].

According to the veterinary students surveyed, most of the antibiotics (93.1% cases) were prescribed by a physician or dentist. However, it should be noted that the remaining respondents declared that they used the antibiotic leftover from the previous therapy, donated by a family member or friend, or purchased at a pharmacy without a prescription. Given the importance of antimicrobial resistance worldwide, irrational use of antibiotics and the prevalence of antibiotic self-medication continue to be significant issues affecting the inappropriate and overuse of this group of drugs. In a 2016 survey of medical, pharmacy, veterinary, dental, and nursing students from 25 UK universities, more than a third (86/242) had taken antibiotics in the past year, of which 3.5% had received antibiotics from friends or family [30]. As indicated by other reports, a significant proportion of medical students also use self-medication with antibiotics in Italy (45%), Kosovo (63.2%), Pakistan (60%), northern Nigeria (38.8%), Australia (91.7%), and Kashmir (80.89%) [31,32,33,34,35,36]. Our study also indicated the declaration that more than half (56%) of veterinary students from senior years (IV–VI) would prescribe themselves an antibiotic *ad usum proprium*.

In addition to the use of self-medication, our study indicates that veterinary medical students show an increased willingness to question prescribed therapy by physicians and treatments recommended by veterinarians. Perhaps this is linked to increasing awareness of existing recommendations for antibiotic use in specific clinical situations in animals. On the other hand, it may also be a symptom of lack of trust in colleagues and lack of humility [23].

The transmission of antimicrobial-resistant bacteria from animals to humans through the food of animal origin is well documented [37]. The first cases of antibiotic resistance in edible animals were reported in 1951 after streptomycin was administered to turkeys [37]. Since then, resistance to antibiotics such as tetracyclines, sulfonamides, β-lactams, and penicillins has been increasingly observed [38,39,40,41,42,43,44,45]. It is somewhat disturbing that more than 1/3 of the respondents in our study still disagree with the statement that the misuse of antibiotics in veterinary medicine causes the phenomenon of antibiotic resistance. Additionally, 2/3 of respondents do not perceive a threat for this issue in the context of possible antibiotic misuse by dentists. Imparting this knowledge to veterinary students seems crucial in forming their attitudes regarding using rational antibiotic therapy in the future.

It is additionally worth noting that one in five respondents (20.8%) disagree with the statement that the overuse of antibiotics by physicians causes the phenomenon of antibiotic resistance. Additionally, two-thirds (66.8%) do not perceive a threat for this issue in the context of possible antibiotic overuse by dentists. Meanwhile, general practitioners and dentists are responsible for nearly 80% of all antibiotic prescriptions written [46,47]. There is an assumption that most of these prescriptions are issued without prior diagnostic tests and, especially among GPs, are for viral conditions for which antibiotics do not work [48].

Students agree that the public is not adequately informed about veterinary aspects of public health provision. Similarly, most students rate the public’s knowledge about antibiotics as very low. It is also puzzling that respondents who were aware of and declared knowledge about the One Health program, the National Antibiotic Program, or European Antibiotic Awareness Day were in the vast minority. In the era of ever-increasing antibiotic resistance of microorganisms, the above facts should persuade relevant institutions (including universities) to pay particular attention to promoting and broadening this knowledge to raise the awareness of the public, including future veterinarians. Promoting One Health attitudes among future veterinarians is especially important due to the emergence of new zoonotic diseases. For this reason, veterinary degree programs should be reformulated under this concept [49].

Given that 99% of respondents expressed a desire to further increase their knowledge of antibiotics, it would be appropriate to revise the scope of curriculum and perhaps expand on topics related to antibiotics and antibiotic resistance. Although evaluating the curriculum was beyond the aim of this study, paying more attention during teaching to issues related to the prudent use of antimicrobials will undoubtedly lead to greater awareness of this topic in future veterinarians. These observations coincide with a recent study that was a joint initiative of the ESCMID Study Group on Veterinary Microbiology (ESGVM) and ESGAP to identify needs and gaps in the antimicrobial stewardship education of European veterinary students [50]. This study revealed a clear students need to broaden knowledge on antimicrobial use, indicating the need to expand European veterinary curricula to include this topic. Australian veterinary students also expressed the need for more education, especially in veterinary pharmacology [51]. Similar requirements were defined in earlier studies conducted on medical students from France and the United States [52,53]. Veterinary medical students from Nigeria indicated a need for better education on antimicrobial resistance and antimicrobial prescribing before graduation and the need for controlled dosing access for over-the-counter drugs [54].

### Limitations of the Study

To fully appreciate the value of the conducted research, some limitations are also worth noting. First, it turns out that—as declared by the respondents to the question about the reasons for choosing the field of study—many people study veterinary medicine by coincidence or because they found it an exciting profession. For this reason, their knowledge of antibiotic use in veterinary medicine, especially in the first years of studies, is more imagined than real. It is especially true when they do not have an animal of their own treated with antibiotics by a veterinarian or their parents do not run a farm-based on animal husbandry. Second, although the students declared a high level of knowledge about antibiotics, compared to the average level of knowledge of the whole society on this topic, their knowledge about antibiotic resistance is much lower than declared. Therefore, it should be assumed that the answers to the knowledge questions do not always correspond to the actual knowledge acquired during the studies. Third, the study showed that the higher the year of veterinary medicine study, the higher the knowledge of antibiotic use and antibiotic resistance. Therefore, it would be expected that students in their final sixth year of study would have the most knowledge. Unfortunately, due to the special nature of this year and die to the epidemiological situation caused by COVID-19, it was difficult to reach an adequate number of students from this year. Sixth year veterinary students should be considered underrepresented in this study.

## 4. Materials and Methods

### 4.1. Study Design and Population

The survey was conducted between May and June 2021. Students from four faculties of academic centers in Poland that run the field of veterinary medicine were invited to participate in the study. These were: Faculty of Veterinary Medicine of the University of Life Sciences in Lublin, Faculty of Veterinary Medicine of the Warsaw University of Life Sciences, Faculty of Veterinary Medicine of the University of Life Sciences in Wrocław, Faculty of Veterinary Medicine of the University of Warmia and Mazury in Olsztyn. The key to selecting universities for the study was that they are located in geographically different parts of Poland and are situated in more and less urbanized regions. They are also faculties of veterinary medicine, which occupy the first four places in the ranking of faculties of veterinary medicine in Poland.

### 4.2. Sample Size

The number of students majoring in veterinary medicine in Poland is approximately 5200. From the selected four centers, the number of students who could potentially participate in the study was 3800. Four hundred and sixty-seven students participated in the study. The study sample was representative of veterinary medicine students in Poland. Considering the size of this sample and the number of veterinary medicine students in Poland (*n* = 5200), the maximum error was 3%, for a confidence level of 95% and a fraction size of 0.5.

### 4.3. The Questionnaire and Data Collection

The questionnaire used in this study is a modified version of a questionnaire created by the authors of a survey of attitudes and knowledge about antibiotics among human medical students [Sobierajski et al., 2021]. The original questionnaire was expanded to include questions about antibiotic therapy and replacement therapy in veterinary medicine and animal husbandry. Changes to the questionnaire were based on the latest knowledge in antibiotic resistance, veterinary medicine, and sociology available to the study’s authors. The questionnaire consists of 4 sections: Demographics, General knowledge, Awareness of functioning programs promoting antibiotic awareness, Knowledge of resistance transmission, and alternative treatments.

Section 1 included questions about gender, year of study, place of origin, or reason for choosing veterinary medicine as a major. Section 2 had queries about taking antibiotics, sources of obtaining antibiotics taken, self-assessment of knowledge about antibiotics, and attitude toward antibiotics. Section 3 included questions about the phenomenon of antibiotic resistance and the reasons for its rise. Section 4 had questions about the rationale behind the use of antibiograms and the reasons for increasing multi-resistant strains in animals. All questions used in the questionnaire were closed-ended questions, including single-choice and multiple-choice questions and questions that used a six-point scale—an extended Likert scale.

The survey questionnaire consisted of 41 questions, and the questionnaire took an average of 30 min to complete. The questionnaire was distributed using a link to the Google Forms application. Four coordinators handled the distribution of the questionnaire at each of the selected universities. The questionnaire was distributed online due to epidemiological restrictions introduced by the Ministry of Health and—resulting from this—the implementation of classes at universities in remote form.

A pilot study preceded the actual survey to verify and evaluate the prepared tool. Nineteen students participated in the pilot study. Pilot participants made some minor comments, which were considered in the final version of the questionnaire.

### 4.4. Statistical Analysis

Cross-tabulations and chi-squared tests evaluated selected factors in antibiotic use and antibiotic-resistance incidence versus student knowledge. All statistical analyses were performed in IBM SPSS Statistics 27.0.1.0. For all analyses, a *p*-level of <0.05 was considered statistically significant.

### 4.5. Ethical Considerations

Consent to conduct the survey among students of Faculty of Veterinary Medicine at the University of Warmia and Mazury in Olsztyn, Faculty of Veterinary Medicine at the Warsaw University of Life Sciences, Faculty of Veterinary Medicine at the University of Life Sciences in Lublin, and Faculty of Veterinary Medicine at the Wroclaw University of Environmental and Life Sciences were also expressed by Deans of the Faculties mentioned above. Study participants were informed that the study was anonymous and confidential, and no personal information, including computer IP, was collected. Results were collected and analyzed collectively to prevent any possibility of identifying the study participant.

## 5. Conclusions

Most of the surveyed veterinary students pointed out that antibiotic resistance is a serious problem. At the same time, a large group still does not think it is a global problem. The students stated that the phenomenon of antibiotic resistance is influenced by overuse and abuse of antibiotics both in veterinary medicine and medicine and animal production, as well as the low awareness of the phenomenon not only by general public but also by professionals’ low hygiene of breeding and animal production contribute to the dissemination antibiotic resistance.

The study suggests that it is necessary to increase the level of antibiotic knowledge of veterinary students especially related to their practical use in their future work as veterinarians, and to pay more attention to universal education on the phenomenon of antibiotic resistance.

## Figures and Tables

**Table 1 antibiotics-11-00115-t001:** Sociodemographic characteristics of study participants (*n* = 467).

		*n* (%)
Gender	Female	358 (76.7)
	Male	102 (21.8)
	Other	7 (1.5)
Year of study	I	78 (16.7)
	II	94 (20.1)
	III	90 (19.3)
	IV	106 (22.7)
	V	69 (14.8)
	VI	30 (6.4)
University location	Lublin	81 (17.4)
	Olsztyn	198 (42.4)
	Warszawa	67 (14.3)
	Wrocław	121 (25.9)
Place of provenance	Big city with more than 200k inhabitants	120 (25.7)
	Medium-sized city with over 50k to 200k inhabitants	90 (19.3)
	Small town up to 50k inhabitants	105 (22.5)
	Village without a farm	92 (19.7)
	Village with a farm	60 (12.8)
Professional interests in the context of future work as a veterinarian *	Companion animals	294 (62.9)
	Livestock/farm animals	95 (20.3)
	Exotic animals	18 (3.9)
	Pharmaceutical industry	37 (7.9)
	Scientific work/laboratory work	86 (18.4)
	Government institution	22 (4.7)
	Do not know yet	78 (16.7)
Do you have a pet?	Yes	426 (91.2)
	No	41 (8.8)
Why did you choose veterinary medicine to study? *	Family reasons (e.g., tradition)	22 (4.7)
	I love animals	250 (53.5)
	This profession seems very interesting to me	343 (73.4)
	Under the influence of friends	1 (0.2)
	I did not get into medical school	52 (11.1)
	Hard to say	38 (8.1)
	Other	23 (4.9)

* Possibility to mark more than one answer.

**Table 2 antibiotics-11-00115-t002:** Use of antibiotic therapy by study participants (*n* = 467).

		*n* (%)
When was the last time you took antibiotics?	Within the last 12 months	164 (35.1)
	1–2 years ago	118 (25.3)
	More than 2 years ago	138 (29.5)
	Never	6 (1.3)
	I do not remember	41 (8.8)
Where did you get the antibiotics you used? **	It was prescribed to me by my family doctor	272 (64.9)
	It was prescribed to me by a doctor of other specialties	82 (19.6)
	It was given to me by a veterinarian	4 (0.9)
	It was prescribed to me by a dentist	36 (8.6)
	It was prescribed to me by a nurse	0 (0)
	I bought it in the pharmacy without prescription	7 (1.7)
	I had an antibiotics at home from a previous treatment	5 (1.2)
	I got the antibiotics from a family member/friend	10 (2.4)
	Other	3 (0.7)
Did you complete a whole course of antibiotic prescribed? **	Yes	392 (93.6)
	No	27 (6.4)
How would you rate your knowledge of antibiotics?	Very bad	2 (0.4)
	Bad	35 (7.5)
	Rather bad	98 (21.0)
	Rather good	184 (39.4)
	Good	115 (24.6)
	Very good	33 (7.1)
	Mean	4.01
	95% IC	3.92–4.11
	SD	1.039

* Possibility to mark more than one answer. ** The number *n* = 419 represents the number of people who said they had ever used an antibiotic and remembered when they used it.

**Table 3 antibiotics-11-00115-t003:** Effect of college curriculum on attitudes toward antibiotic use and the phenomenon of antibiotic resistance (*n* = 467).

		Self-Assessment of Respondents’ Knowledge about Antibiotics * (1—Very Bad, 2—Bad, 3—Rather Bad, 4—Rather Good, 5—Good, 6—Very Good)		
	Total *n* (%)	1, 2, 3 * *n* (%)	4, 5, 6 * *n* (%)	*p*-Value
**Have you ever been taught about the growing problem of antibiotic resistance during your studies?**				<0.001
**Yes**	418 (89.5)	109 (26.1)	309 (73.9)	
**No**	49 (10.5)	26 (53.1)	23 (46.9)	
**Did your veterinary college classes influence you to gain more knowledge about the use of antibiotics in humans?**				<0.001
**Yes**	325 (69.6)	67 (20.6)	258 (79.4)	
**No**	142 (30.1)	68 (47.9)	74 (52.1)	
**Have your veterinary college classes influenced you to gain more knowledge about the use of antibiotics in animals?**				<0.001
**Yes**	375 (80.3)	81 (21.6)	294 (78.4)	
**No**	92 (19.7)	54 (58.7)	38 (41.3)	
**Did the knowledge you gained about antibiotics in your veterinary studies influence you to negate the therapy ordered by your doctor for your disease?**				
**Yes**	113 (24.2)	28 (24.8)	85 (75.2)	0.008
**No**	354 (75.8)	107 (30.2)	247 (69.8)	
**Did the knowledge of antibiotics you gained in veterinary college impact negating the therapy ordered by a veterinarian for your or your friends’ pet’s illness?**				
**Yes**	115 (24.6)	26 (22.6)	89 (77.4)	0.003
**No**	352 (75.4)	109 (31)	243 (69)	

* Type of Likert scale was used in the question about self-assessment knowledge about antibiotics: 1—very bad, 2—bad, 3—rather bad, 4—rather good, 5—good, 6—very good.

**Table 4 antibiotics-11-00115-t004:** Use of antibiotics by veterinarians for personal use (*n* = 467).

		Self-Assessment of Respondents’ Knowledge about Antibiotics * (1—Very Bad, 2—Bad, 3—Rather Bad, 4—Rather Good, 5—Good, 6—Very Good)		
	Total *n* (%)	1, 2, 3 * *n* (%)	4, 5, 6 * *n* (%)	*p*-Value
**Have you ever encountered a situation where a veterinarian prescribes an antibiotics for their use (*ad usum prioprium*), that is, they write the drug for himself/themselves and take it himself/themselves?**				
**Yes**	103 (22.1)	23 (21.4)	80 (78.6)	0.046
**No**	364 (77.9)	112 (30.8)	252 (69.2)	
**As a veterinarian, would you prescribe yourself an antibiotics for personal use (*ad usum prioprium*)?**				
**Yes**	242 (51.8)	60 (24.8)	182 (75.2)	0.019
**No**	225 (48.2)	75 (33.3)	150 (66.7)	

* Type of Likert scale was used in the question about self-assessment knowledge about antibiotics: 1—very bad, 2—bad, 3—rather bad, 4—rather good, 5—good, 6—very good.

**Table 5 antibiotics-11-00115-t005:** Practical knowledge of antibiotics (*n* = 467).

To What Extent Do You Agree or Disagree with the Following Statements?	Median *	25th–75th Percentiles *
Antibiotics are effective against viruses	1	1-1
Antibiotics are effective against bacteria	6	5-6
Improper use of antibiotics can cause microorganisms to become resistant	6	6-6
Using antibiotics will make people resistant to them	2	1-3
The use of antibiotics often causes side effects (e.g., diarrhea, headaches, stomach pain, allergies)	4	3-5
Antibiotics are effective for the common cold	2	1-2
Doctors often prescribe an antibiotic unnecessarily	4	3-5
Bacteria pass information to each other about antibiotic resistance	6	4-6

* Median and 25th–75th Percentiles on a scale of 1 to 6, where 1 meant “strongly disagree” and 6 meant “strongly agree”.

**Table 6 antibiotics-11-00115-t006:** Assessing the influence of factors on the emergence of antibiotic resistance (*n* = 467).

Factors, Attitudes, Behaviors *	Strongly Disagree	Disagree	Rather Disagree	Rather Agree	Agree	Strongly Agree
	*n* (%)	*n* (%)	*n* (%)	*n* (%)	*n* (%)	*n* (%)
**Overuse of antibiotics by doctors**	9 (1.9)	20 (4.3)	68 (14.6)	68 (14.6)	120 (25.6)	182 (39)
**Overuse of antibiotics by dentists**	62 (13.3)	95 (20.3)	155 (33.2)	84 (18)	33 (7.1)	38 (8.1)
**Use of antibiotics without a prescription (self-medication)**	77 (16.5)	56 (12)	63 (13.5)	52 (11.1)	80 (17.1)	139 (29.8)
**Misuse of antibiotics in veterinary medicine**	20 (4.3)	59 (12.6)	103 (22.1)	75 (16.1)	119 (25.5)	91 (19.5)
**Low awareness of the dangers of antibiotic resistance**	5 (1.1)	13 (2.8)	44 (9.4)	40 (8.6)	130 (27.8)	235 (50.3)
**Limited access to microbiological diagnostics**	19 (4.1)	45 (9.6)	85 (18.2)	94 (20.1)	111 (23.8)	113 (24.2)
**Use of antibiotics in fattening breeding animals**	33 (7.1)	46 (9.9)	84 (18)	98 (21)	97 (20.8)	109 (23.3)
**Too long antibiotic therapy in animals**	45 (9.6)	66 (14.1)	136 (29.1)	114 (24.4)	61 (13.1)	45 (9.6)
**Using too low doses of antibiotics in animals**	47 (10.1)	83 (17.8)	138 (29.6)	91 (19.5)	59 (12.6)	49 (10.5)
**Low level of hygiene in animal husbandry**	14 (3)	54 (11.6)	85 (18.2)	71 (15.2)	107 (22.9)	136 (29.1)

* Q: Do the following factors/attitudes/behaviors contribute to the rise of antibiotic resistance?

## Data Availability

The dataset used during this study is the available form given author upon reasonable request.

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
