# Peer review of "Antimicrobial and Antibiotic Resistance from the Perspective of Polish Veterinary Students: An Inter-University Study"

_antibiotics, 2022, doi:10.3390/antibiotics11010115_

Round 1
Reviewer 1 Report
This is an interesting study on an important topic such as microbial resistance. I only have few comments for the authors:
- add study design in the title
- add numbers to the abstract and p values
- line 41 - vice versa in italic
- line 47 who abbreviation not introduced before
- Table 1 please divide in 2 tables; the second one should start with When was the last time you took atb
- Table 4 - results should be expressed as median and IQR
- line 348 - why is considered in smaller font
- use Kruskal-Wallis test for likert results
Author Response
Thank you very much for your excellent evaluation of our research paper and the article prepared based on it.
We also thank you for your very insightful analysis and for pointing out the places that needed improvement. We have considered them all.
Thank you for your suggestions regarding the analysis and content of the text - we have also mostly considered them.
- add study design in the title
We specified that this is study on polish students of veterinary
- add numbers to the abstract and p values
We added data on the global problem of antibiotic resistance
- line 41 - vice versa in italic
We corrected that
- line 47 who abbreviation not introduced before
We corrected that
- Table 1 please divide in 2 tables; the second one should start with When was the last time you took atb
We changed that
- Table 4 - results should be expressed as median and IQR
We changed that
- line 348 - why is considered in smaller font
We corrected it
- use Kruskal-Wallis test for likert results
We will use this excellent indication for the following research publication when the responses to the Likert scale questions will be analyzed in more detail. We analyzed some data with the Kraskal-Wallis test, but we decided not to include them in this paper.

Reviewer 2 Report
Title: add "Polish", in "...Perspective of Polish Veterinary Students."
Lines 245-247:
1.
a. it is important to clarify what was the goal of the study - was it to evaluate the quality of the vet studies/curriculum in Poland, or
b. to evaluate the AMR knowledge/awareness among vet students?
They are related, but not the same.
2. What is stated in these lines, seems to be contradicated by several of the findings of the study, when a significant number of the students expressed AMR gaps
Author Response
Thank you very much for your excellent evaluation of our research paper and the article prepared based on it.
We also thank you for your very insightful analysis and for pointing out the places that needed improvement. We have considered them all.
Thank you for your suggestions regarding the analysis and content of the text - we have also mostly considered them.
- Title: add "Polish", in "...Perspective of Polish Veterinary Students."
We added an adjective in the title of the article and
- Lines 245-247:
- it is important to clarify what was the goal of the study - was it to evaluate the quality of the vet studies/curriculum in Poland, or
- to evaluate the AMR knowledge/awareness among vet students?
They are related, but not the same.
We explained in the Abstract what the purpose of the study was.
- What is stated in these lines, seems to be contradicated by several of the findings of the study, when a significant number of the students expressed AMR gaps
In the Discussion, we completed the description of the data so that everything was understandable(Lines: 251-255)
“It also allows us to conclude that the course of studies conducted at the faculties of veterinary medicine in Poland reports the phenomenon of antibiotic resistance quite reasonably. However, there are still many elements that would need to be strengthened in the learning process, with particular attention to the global nature of AMR.”

Reviewer 3 Report
The study addresses a very important topic that is currently of great interest. The text is clear, objective and well-written. We suggest your acceptance with minor corrections. The suggestions are described in the pdf file.

Author Response
Thank you very much for your excellent evaluation of our research paper and the article prepared based on it.
We also thank you for your very insightful analysis and for pointing out the places that needed improvement. We have considered them all.
Thank you for your suggestions regarding the analysis and content of the text - we have also mostly considered them.
- Is not clear that this number is the scale, please describe better.
We explained and wrote what the numbers used in the tables are (Line 110 and Line 139)
- Please include the data description in the indicator column of the table or chart.
We corrected that
